# Assessment of Community Behavior and COVID-19 Transmission during Festivities in India: A Qualitative Synthesis through a Media Scanning Technique

**DOI:** 10.3390/ijerph191610157

**Published:** 2022-08-16

**Authors:** Sumit Aggarwal, Nupur Mahajan, Simran Kohli, Sivaraman Balaji, Tanvi Singh, Geetha R. Menon, Kiran Rade, Samiran Panda

**Affiliations:** 1Division of Epidemiology and Communicable Diseases, Indian Council of Medical Research, V. Ramalingaswami Building, Ansari Nagar, New Delhi 110029, India; 2Indian Council of Medical Research-National Institute of Medical Statistics, New Delhi 110029, India; 3Central Tuberculosis Division, World Health Organization, New Delhi 110002, India

**Keywords:** COVID-19, festivals, COVID-appropriate behavior, media scanning, India

## Abstract

In India during the first wave of COVID-19 infection, the authorities were concerned about the advent of the festive season, which could lead to a surge in cases of SARS-CoV-2 infection. The present study attempted to assess the socio-behavioral aspects of COVID-appropriate behavior (CAB) at individual and community levels, and their impact on the transmission of COVID-19 during festivities in India. Media scanning was conducted to qualitatively assess CAB by analyzing 284 news reports from across India; quantitative data on COVID-19 daily cases from March 2020 to December 2020 were used to determine the trends of the adjusted test positivity (ATP) ratio for six cities. Peaks in ATP were observed in Chandigarh, Delhi-NCR in North India during and after Dussehra and Deepavali, and in Mumbai, in the west, after Navratri. Additionally, a surge in ATP was observed in Trivandrum after Onam and in Chennai around Deepavali in the south; meanwhile, in the eastern city of Kolkata, cases increased following Durga Puja. The major challenges were adherence to CAB viz. social distancing, hygiene, and compliance with the mask mandate. Microlevel CAB indicated relatively higher laxity in maintaining hand hygiene in all cities. Observations from the current study indicate that innovative community-driven initiatives during festivals in each geographical zone are key to the large-scale implementation of disease prevention measures.

## 1. Introduction

Since December 2019, the coronavirus disease 2019 (COVID-19) has impacted every aspect of human life in unfathomable ways around the globe [1]. Numerous countries took unprecedented measures to prevent the spread of the disease, such as closing schools, restaurants, and places for social gatherings, prohibiting religious congregation, and restricting the usage of public transport [2]. The pandemic thus drastically changed the nature of social interactions [3]. In India, a nationwide lockdown was imposed from 25 March 2020 until 31 May 2020, to curb the spread of SARS-CoV-2—the virus that causes COVID-19. Restrictions on intra- and inter-state travel, provisions for home working, and the closure of educational institutions and religious and public places, along with a ban on public gatherings for festivities or other occasions, were also imposed [4].

Past studies have shown that the transmission of infectious diseases increases following mass gatherings during festivals, congregations, and large-scale musical or sporting events [5,6,7,8,9]. Notably, festivals are staged in various locations, both indoors and outdoors, and are managed both by public sector services and by privately run organizations [1,10]. With the advent of COVID-19, peaks in cases were distinctly visible globally after sudden influxes of people owing to festivities, especially after lockdown restrictions were eased. Several reports from the United States, Italy, the United Kingdom, Bangladesh, and also from several states of India, particularly in locations where annual gatherings occurred as part of traditions or recreation, indicated surges in cases after festivals [11,12,13,14].

In the current context, the Government of India has coordinated with state governments to implement guidelines for low-key celebrations during religious festivals such as Ramadan, Eid, Ram Navami (the festival commemorating the birth of Lord Rama), Ganesh Utsav (celebrations related to Lord Ganesha), Navratri (a week-long festival worshipping Goddess Durga), and Deepavali (the festival of lights) during phases of lockdown and subsequent unrestricted periods. However, there have been instances of laxity among people after restrictions were relaxed during the phased unlocking of the country and the states [15,16] (Sinha et al., 2020; Taskin 2020). Later in 2021, mathematical-modelling-based projection was used to formulate guidelines for populated tourist destinations, and to encourage responsible rather than revenge travelling [17] (Mandal et al., 2021). The festive season that commenced in mid-October 2020 saw reports of COVID-appropriate behavior (CAB) being flouted in densely populated cities, with daily reporting of an average of 5000 cases for four consecutive weeks [18] (Ministry of Health and Family Welfare, 2020). Owing to the observed spike in cases with the advent of India’s festive season, the Indian government launched “Jan Andolan”, which has the colloquial meaning of “the people’s campaign for CAB” [19]. Despite such approaches and guidelines, the festive season triggered an increase in the movement of people for shopping, leisurely visits, and ceremonial get-togethers, thereby enhancing the chance of infection [13,19,20]. In this context, we aimed to explore different social-behavioral factors and their plausible effect on COVID-19 transmission during festive seasons in India.

## 2. Materials and Methods

### 2.1. Data Source

The authors’ institute, which is situated in India, has maintained a real-time online server to document the number of COVID-19 tests conducted in India, and their outcomes, since March 2020. District-wise testing and positivity data were collected from this server during the period from March 2020 to mid-December 2020. The timeline for data collection related to festivals was the same as that for the COVID-19 test data.

#### 2.1.1. Screening of Festivals and the Media Scanning Methodology

The list of major festivals celebrated in India was collected from the website of the Ministry of Culture, Government of India, in October 2020 (Appendix A
Table A1). In order to identify local festivals, state websites and state calendars were explored. Community behavior during festival season in India was assessed by ‘media scanning’, as information related to CAB directly observed by people was sparse. This approach provided a community-focused outlook, which is sometimes overlooked in simple scientific literature searches. Moreover, reports from the local and national daily newspapers and online portals provided a comprehensive picture to the researchers and readers of the community behavior at large.

#### 2.1.2. News Article Search Strategy

A detailed search strategy with inclusion and exclusion criteria was devised, whereby online news articles from selected publication houses, Google news, and other open-source news platforms were screened. Keywords such as festival name AND “Corona” OR “Coronavirus” OR “COVID-19” AND COVID AND India AND 2020 were used to scan the newspaper articles and videos through online portals. The top 10 most read/circulated English and Hindi daily newspapers in India, as identified by the Office of the Registrar of Newspapers of India (2020), were scanned through the three most common search engines: Google, Yahoo, and Bing [21]. Additionally, media scanning was also conducted in regional languages, such as Kannada, Malayalam, Tamil, Odia, and Bangla.

##### Inclusion and Exclusion Criteria for Media Scanning

Articles that reported COVID-19 cases and highlighted CAB among population groups in various districts and cities of India were shortlisted and selected for qualitative data analysis. Articles that presented duplicate information in multiple languages, those articles not linked to the study objectives, and articles providing irrelevant and incomplete information were excluded from the study. A total of 756 news reports on 16 festivals and COVID-19 were collected through media scanning. Of these, 284 were found suitable for content analysis. (Figure 1).

Although the reporting of the events by the newspapers was taken at face value, individual experiences of the members of the research team conducting the current investigation offered an additional information base for understanding COVID-appropriate behavior at various event sites. Meanwhile, the details regarding the festivals, which were taken from the Government website, did not require any verification.

#### 2.1.3. Guidelines Related to Lockdown

The Government of India imposed a full national-level lockdown from the last week of March to the end of May 2020, following which unlocking was commenced in a phased manner. The Ministry of Home Affairs declared the timings of these restrictions, and also issued guidelines and regulations pertaining to mass gatherings and public activities during lockdown and unlocking [22]. In order to understand government guidelines, these regulatory documents were collected and analyzed. The activities that were permitted and restricted, during both complete lockdown and unlocking, are presented in Figure 2.

### 2.2. Data Analysis

#### 2.2.1. Identification of Behavior Domains and Cross-Verification

The selected domains were reflective of various CABs adopted at the individual and community levels to facilitate the prevention of the spread of SARS-CoV-2.

During COVID, there were numerous media reports across the country on the spread of infections following various public gatherings. Sometimes, the same event was covered multiple times across different newspapers. The team initially collected all these reports (*n* = 756) and then, after carefully applying the exclusion and inclusion criteria, 284 articles were deemed fit for content analysis. From these articles, 25–30 articles were randomly selected to identify the domains that would ensure representation of diversity both in the population covered and in the media sources. The broad themes for the larger analysis were kept in mind while extracting the domains. The final analysis was, however, conducted on all of the included articles based on the extracted domains. The analysis was carefully conducted by public health experts, anthropologists, and psychologists to ensure the relevance and feasibility of continuing with the content analysis.

#### 2.2.2. Content Analysis

The content analysis technique was utilized to determine the presence of CAB-related key words, themes, or concepts in the newspaper reports. These were analyzed to identify the presence, meanings, and relationships of these contents, as per the study objectives.

Content analysis of media-scanned reports was carried out, and behavior was quantified for the occurrence/absence of the domains in the form of frequencies. This was carried out for each domain and its respective sub-domain by two independent researchers. For the final frequencies, inter-rater agreement of the independently assigned values was estimated and discrepancies were resolved by going through the article together. Such parameters were also examined to assess the extent of implementation of CAB by the authorities in the selected cities from the north, west, south and east zones of India. Furthermore, the reported number of tests that were conducted and returned a positive result was used for estimation of the adjusted test positivity ratio (TPR). Trendlines were plotted for selected cities to assess the relation between peaks of adjusted TPR, and peri-festive and festive activities.

#### 2.2.3. Estimating the Adjusted Test Positivity Ratio

TPR, which is the ratio of the number of positive test results and the number of tests performed, may be misinterpreted, as both the numerator and the denominator were changing due to various reasons, such as the scaling up of testing capacity, changes in testing criteria for COVID-19, and changes in the number of cases at the beginning and later over a period of time [23]. Therefore, adjusted TPR was estimated to assess the trends of test positivity in selected cities. It was also used to determine whether festive seasons and associated movements of people influenced test positivity in the days following these events.

Adjusted TPR on the day ‘t’ was calculated by multiplying the observed TPR with the daily ratio of increase in cases to tests [24].

Adjusted TPR = observed TPR * z_t_, where z_t_ = r_case_t_/r_test_t_, and where r_case_t_ = C_t_ − (C_t–1_)/C_t–1_ is the growth rate of cases, and r_test t = T_t_ − (T_t–1_)/T_t –1_ is the growth rate for tests.

C is the reported number of cases on day t or t−1; T is the reported number of tests conducted on day t or t−1.

## 3. Results

The geographical analysis of the shortlisted articles showed that news reports covered 64 geographical locations from various states in India (Figure 3), which were funneled into six cities to assess the trend of ATPR and to examine whether peri-festive activities and festivals were temporarily associated with the caseload. Details of the number of articles analyzed for each of the reported festivals are provided in Appendix A
Table A1.

### 3.1. Identification of Challenges and Innovations

Content analysis of the scanned reports indicates that the challenges faced by authorities and community-based-event organizers were overcome by the adoption of innovative strategies during festivals. Figure 4 illustrates the challenges and innovations identified during media scanning of reports. A total of eight domains reflecting COVID-19 prevention-related behaviors were identified; those not fairly representing the behavior, and a few related domains which were indicative of the clustering of similar behaviors, were removed.

A multi-level approach was followed for the analysis, which involved forming clusters by dividing the identified domains across two levels. The categorization into these levels was based on whether the domains were individual-centric or operated at a community level. Each domain was further divided into the following behavior sub-domains:Micro parameters to assess individual-centric COVID prevention behavior. Content analysis showed that three individual-level CABs were reported during various festive and peri-festive events to prevent the spread of SARS-CoV-2 infection. Various guidelines from the Government of India also highlighted three individual-centric CABs to lower the risk of disease transmission, as follows:(a)Compliance with proper mask application. The first domain was identified as adherence to the mask mandate, which included the mindful use of face masks by individuals in public spaces. Shopping malls and event venues had implemented ‘no mask, no entry’ policies, while mask distribution drives were organized on various occasions. The use of face masks by individuals in public areas was observed to be followed appropriately across Indian cities, as reported by 61% of media articles.(b)Hand hygiene. News reports captured usage of hand sanitizers, the maintenance of hand hygiene on an individual level, and the installation of sanitizing measures at public events. A lack of compliance with proper hand hygiene among individuals across India was reported in 59% of the news articles analyzed in this study.(c)Practicing social distancing measures. As per the news reports, individuals cautiously avoided overcrowded public spaces. In order to handle overcrowding and problems related to ventilation in event venues, certain innovations were also put in place, such as drive-through and/or four-side-open pandals (temporary sheds for worshipping idols), and the demarcation of spaces to maintain safe distances between individuals. Through content analysis of visual materials, it was reported in 60% of news reports that individuals followed social distancing during pre-festive and festive events in most situations. However, some events witnessed the flouting of guidelines related to social gatherings, as reported in 40% of the reports.Macro parameters to assess community-centric COVID prevention behavior. Residential welfare associations (RWAs) and local authorities chose to develop innovative venues for festive events to prevent the spread of COVID-19. A total of five domains were identified to assess community-based (macro-level) CABs as reflected through media reports across various cities.(a)Mass/community disinfection measures. The first domain was taken as community-level adoption of disinfection measures. A lack of proper disinfection measures was reported in 72.5% of the media articles. However, certain innovative precautionary measures, such as installing sanitizer dispensing machines or sanitization tunnels at the venues of events, and time-to-time sanitization of idols, pandals and event premises, were also reported.(b)Temperature screening measures. Thermal screening of individuals in public places was one of the most common community adopted preventive measure adopted across the country. However, 78% articles reported the absence or improper implementation of this measure in festive celebration venues.(c)Appropriateness of premises. About a third of the news reports highlighted events being organized in open spaces to avoid crowding and poor ventilation. However, adherence to these guidelines to avoid overcrowding at celebration events varied geographically. Interestingly, a similar proportion of news reports indicated that most individuals were celebrating festivals within their residential premises and respective houses with families in a restricted fashion. Local community administrations conducted small-scale celebrations as per the government guidelines in all zones, mainly during Ganesh Utsav, followed by Durga Pooja (the festival for worship of Goddess Durga) and Dussehra (the festival involving the burning of effigies of Ravana, symbolizing Lord Rama’s victory over him), which were telecast on media platforms to keep people at home and experience the festive atmosphere from indoors. In one innovative approach adopted in Kolkata, drive-through viewings of idols and pandals were organized during Durga Pooja. Twenty seven percent of the media-scanned reports indicated that preventive guidelines were flouted.(d)Gathering as per guidelines. Through stringent guidelines, state governments restricted public gatherings in large numbers through different phases of unlocking to avoid a surge in COVID-19 cases. As a result, it was observed that all the regions reported a considerable number of cancellations of public events (34%).(e)Virtual modes of viewing holy idols in the pandals (darshan) and the live-streaming of festival-related events were reported in 18% of the media reports. This kind of innovation has been described as a networked event, and was reported primarily from Mumbai and Ahmedabad during Ganesh Utsav and Krishna Janmashtami (the festival celebrating the birth of Lord Krishna). About a fourth of the reports highlighted breaches of restricted gathering guidelines from several cities.(f)Crowd management. Print media and visual aids reported on crowd management by police, communities, and local community-based organizational associations. However, non-cooperative behavior and flouting of the guidelines by individuals were also reported. Laxity around retail shopping and recreational and religious travel was reported in 32% of the media reports.(g)Notably, a few examples of good crowd management during festivities were identified. Two such events were the Ram Janmabhoomi foundation-stone-laying ceremony in Ayodhya, Uttar Pradesh, and Jagannath Puri Rath Yatra (the Chariot of Lord Jagannath) in Puri, Odisha, in August 2020. According to the reports, management and local authorities allowed limited attendance to those with COVID-19-negative test results. Furthermore, in October 2020, specific measures were endorsed for maximum attention and usage by individuals and communities, especially during the festive season as part of the Jan Andolan campaign. During Mysore Dussehra, approximately 200 people gathered at the Mysore Palace; however, the duration of the cultural programs and the number of artists were reduced considerably; these restrictions were accompanied by the observation of proper sanitization measures.

### 3.2. Trend of ATPR during Major Festivals and the Peri-Festival Season

The trendline for daily ATPR along with festivals from March 2020 to December 2020 for the north, west, south, and east zones of India is depicted in Figure 5a–f. The vertical bars highlight the days of festive events for which media-scanned reports were found in each of the six city clusters. In the present study, it was observed that, from the end of March 2020 to mid-December 2020, there was an increase in daily ATPR after a festive event in that region. In particular, surge peaks in ATPR were observed in Chandigarh, Delhi-NCR, Mumbai, Trivandrum, Chennai, and Kolkata around mid-October 2020. This trend continued until the end of November 2020: the major festive season in the country.

## 4. Discussions

The easing of the national lockdown in India from June 2020 was followed by an increase in the number of new cases of COVID-19 until September 2020, a time marked by the commencement of the festival season in India. The rise in cases during the period of Deepavali in Delhi-NCR and Chandigarh was comparable to the festival-related rises in Kerala, Karnataka, and Telangana during Onam (New Year celebrations among certain communities in South India) [25]. Noticeably, footfall for recreation and retail purchases was at 58% of the pre-pandemic level at the end of September 2020. However, by mid-October 2020, with the commencement of Dussehra, Navratri, and Durga Pooja across the country, there was a surge in mobility of 69%, which had further soared to 76% by Deepavali [26]. This increase in the movement of people within and outside their residential territories during the festive season and peri-festival period, starting weeks before the actual festival date, was characterized by visits to market places for shopping, meeting friends and families, and gift purchases and exchanges, and attendance at fairs, places of worship, and social events [27].

Social distancing, crowd management, the maintenance of hand hygiene, and compliance to mask discipline, as per state-guidelines, were issues that required adherence. Thus, innovations were introduced to achieve an essence of the festive season, while maintaining precautionary and preventive measures. The present study highlighted the innovations administered at individual and community levels to overcome the challenges in managing caseloads of COVID-19 during the festive season [28,29,30,31,32,33]. Innovative drive-through pandals were established in Kolkata during Durga Puja; similar innovations were introduced in other parts of the world, such as the UK and Europe, where drive-in concerts and film screenings were attended by audiences from the safety of their own cars [12].

It was observed that community-driven initiatives were implemented during festivals to control the spread of SARS-CoV-2 infection, and crowd management efforts were taken by the local police. Despite this, these interventions were less successful in populated market areas, and non-compliance to proper masking featured in media reports [16,32,34,35,36]. Similar observations were made regarding mass gatherings in the Burriana town of Spain during the Falles festival, which resulted in the transmission of COVID-19. Participation in the Falles festival was extensive, since the population of Borriana, both adults and children, took part in many cultural, touristic, leisure, and dinner events during the months of February and March 2020. These events were considered to be super-spreader events [37].

The trend of rising ATPR was reported in cities including Chandigarh, Delhi, Mumbai, Chennai, and Kolkata, where non-adherence to micro and macro-level CABs was evident in the media scanning [36,38,39,40]. Similar findings were reported by the State Bank of India (2020), where 50–60% increases in COVID-19 cases were observed after Ganesh Utsav and Onam in cities such as Andhra Pradesh, Maharashtra, Telangana, and Kerala from August 2020 [41]. Several reports mentioned that the surge in cases could be due to multifactorial influences, such as increased levels of pollution, various socio-demographic factors, the emergence of infectious variants, and low vaccine coverage [42,43]. However, travel during festive events, accompanied by unrestricted social interactions, were plausible reasons behind increased test positivity, a theory that later received support from mathematical-modelling-based projections [17]. Thus, there was an evident tension between efforts to curb the spread of the SARS-CoV-2 infection through innovative measures, and the flouting of COVID-19 risk reduction strategies.

### Strengths and Limitations

The present study adds knowledge to the literature on festival research from India, and also provides significant input for the reconstruction of communities and places as the world gradually attempts to come to terms with the grip of the COVID-19 pandemic. This study was based on the analysis of newspaper reports, which the authors juxtaposed against trendlines of adjusted TPR in order to demonstrate a temporal association between the qualitative findings of content analysis and subsequent surges in SARS-CoV-2 infections. Secondly, in this investigation, a media scanning approach was adopted for data collection, as it provided a unique outlook on community behavior, which might have been overlooked in a study that focused on the scientific literature. It is also noteworthy that this article takes into consideration the festive seasons from all zones of India, with reports from local and national daily newspapers and online portals, and therefore provides a comprehensive picture about a less researched area pertaining to COVID-19 transmission. Further, linking reported observations on CAB in newspapers with COVID cases in an area is challenging, and finding such associations does not necessarily mean causation; however, there are no primary studies which objectively record CAB in India in a scientific manner. Overall, this inquiry provides insights into the relationship between festivities, behavior, and disease transmission.

Our study had the following limitations: (i) media scanning was conducted to acquire documents that might be inclusive of the entire dataset, as some newspaper portals require subscriptions; elsewhere, for some news platforms, full texts of back-dated articles were not available, especially for regional sources, (ii) reports on COVID-appropriate behaviors, i.e., hand hygiene practices, the presence of sanitizing measures, and social distancing, were based on observations through photographic evidence, as very few reports quantified them textually, and (iii) due to the discontinuous nature of qualitative data, the association between qualitative domains and daily ATPR was explored through trend analysis and the juxtaposition of event lines following the mixed-method approach.

## 5. Conclusions

Globally, the COVID-19 pandemic demanded changes in the social fabrics of nations. One of its major preventive measures, ‘social distancing’, was considered a forceful control for contagion, but was antithetical to festivals. The trend of rising cases during festivals in India can therefore be attributed to lower adherence to COVID preventive behaviors, such as social distancing, and inappropriate mask usage, along with several other potential contributing factors, including mass gathering events that increased social mixing. Thus, the fear associated with the disease resurfaced with the second wave in India in the first half of 2021, making it even more necessary to adopt CABs. We highlighted the strategies and protocols involving individuals and communities in curbing the caseload of COVID-19 in general and during festive seasons; these findings could inform future pandemic management planning.

## Figures and Tables

**Figure 1 ijerph-19-10157-f001:**
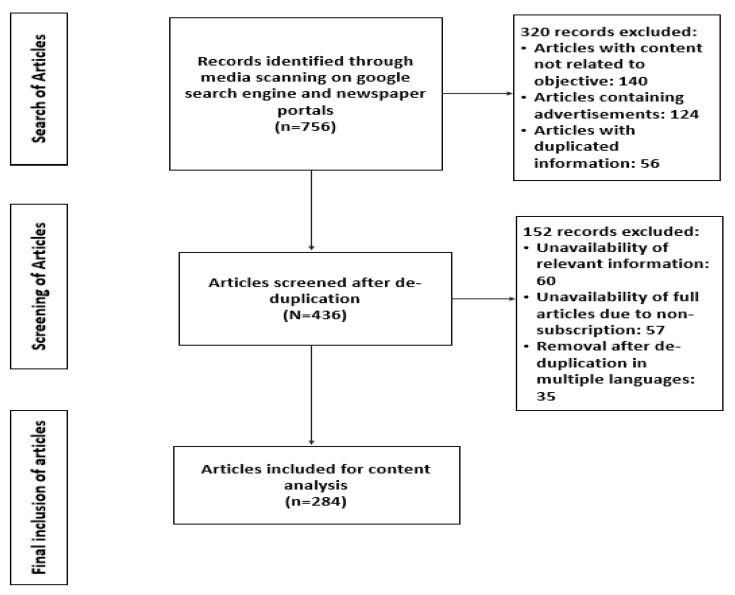
Search strategy and output.

**Figure 2 ijerph-19-10157-f002:**
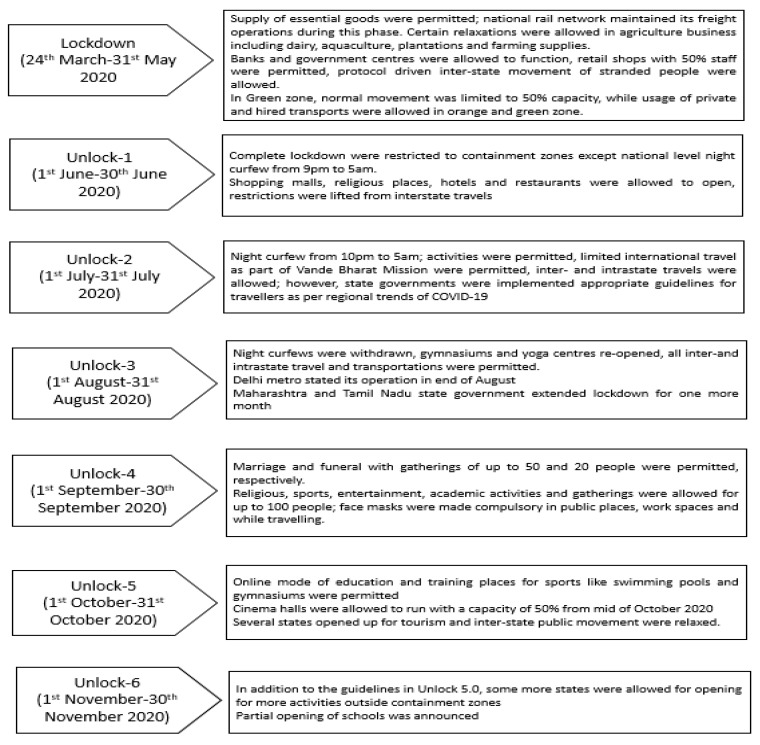
Timeline of lockdown and unlocking phases in India, and the activities permitted during these phases [22].

**Figure 3 ijerph-19-10157-f003:**
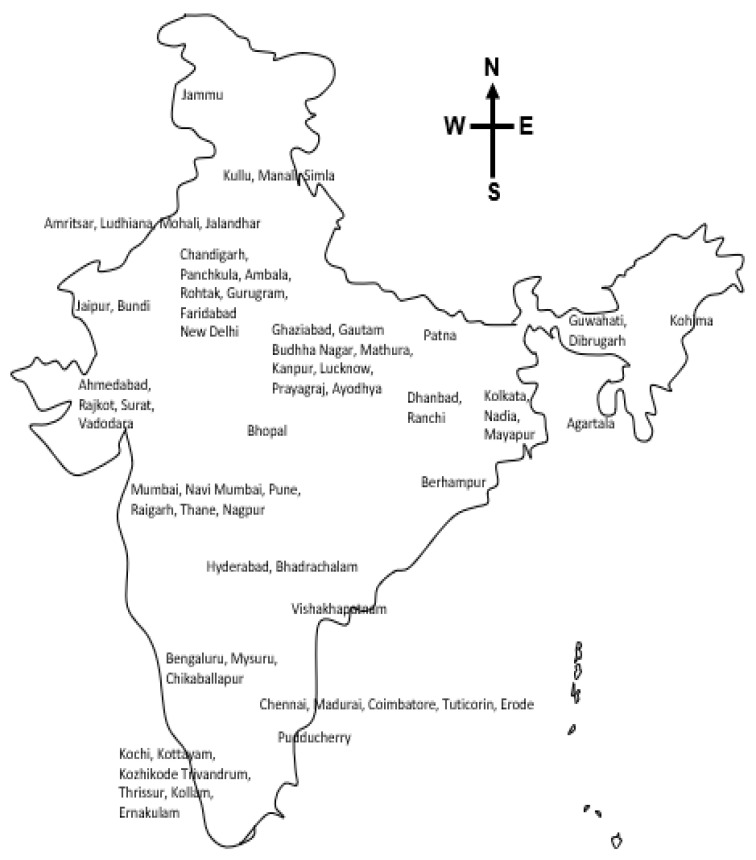
Locations across India contributing media-scanned reports for content analysis.

**Figure 4 ijerph-19-10157-f004:**
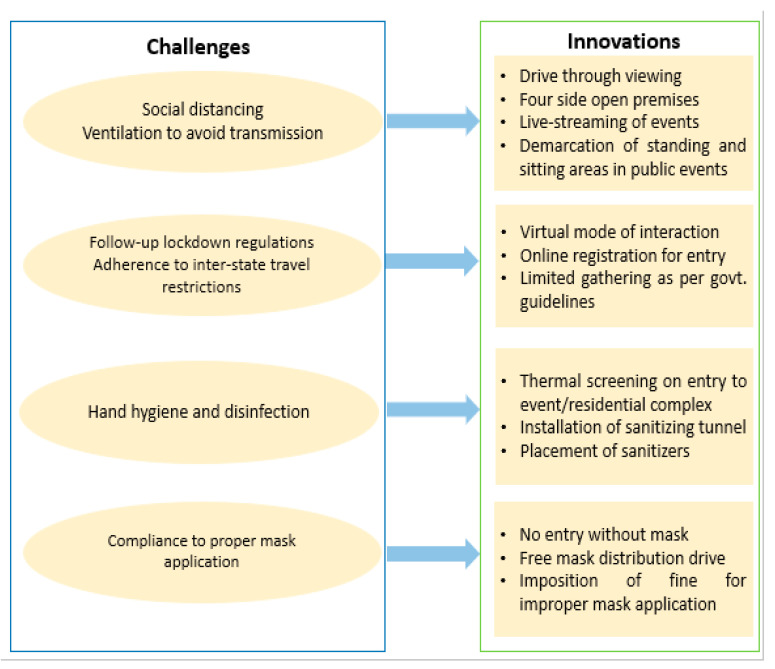
Thematic analysis of challenges and innovations during festive seasons.

**Figure 5 ijerph-19-10157-f005:**
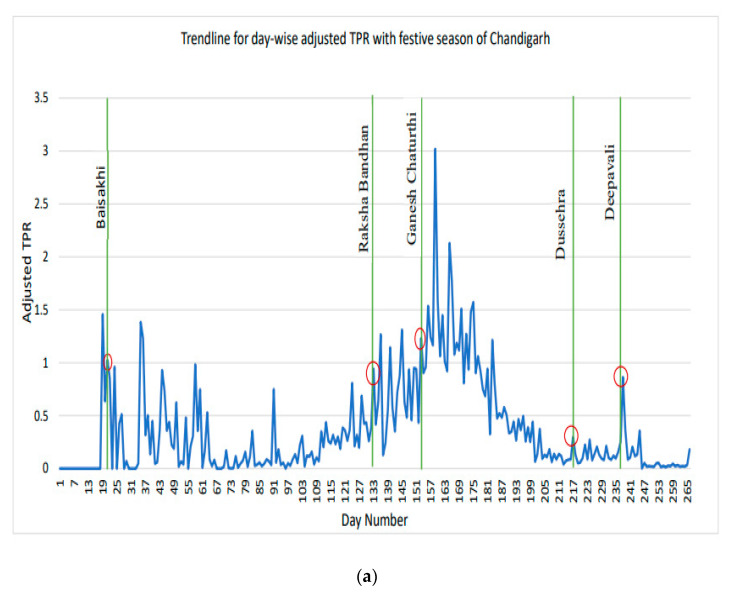
(**a**): Trendline for day-wise adjusted TPR with festive season of Chandigarh. (**b**): Trendline for day-wise adjusted TPR with festive season of Delhi/NCR. (**c**): Trendline for day-wise adjusted TPR with festive season of Mumbai. (**d**): Trendline for day-wise adjusted TPR with festive season of Trivandrum. (**e**): Trendline for day-wise adjusted TPR with festive season in Trivandrum. (**f**): Trendline for day-wise adjusted TPR with festive season in Kolkata.

## Data Availability

The datasets used and/or analyzed in the current study are available from the corresponding author on reasonable request.

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
