# Peer review of "Assessment of Community Behavior and COVID-19 Transmission during Festivities in India: A Qualitative Synthesis through a Media Scanning Technique"

_ijerph, 2022, doi:10.3390/ijerph191610157_

Round 1

Reviewer 1 Report

The paper discusses an influence of the festive season to the transmission of COVID-19 in terms of socio-behavioral aspects of COVID-appropriate behavior. The reported study is relevant to the current pandemic situation and the paper is well-written. My comments are as follows:

- How exactly the information about events was extracted from media sources? How it was verified?
- "25-30 randomly selected media-scanned articles and government guidelines were analyzed" - please, clarify the analysis procedures. Why was random selection used instead of some relevance-based approach?
- If this Referee understands right, you try to unveal the dependencies between mentioning CAB in articles and COVID cases? If yes, this approach does not reveal the dependence between real CAB and its mentioning (e.g. recommendations to keep social distance). What is the meaning of such analysis?
- How do you differ the influence of restrictions and recommendations on CAB?
- How does Fig. 3 support the other results of the study?
- The various covid strains possess different  incubation periods and different contagiousness. How can this be accounted in analysing Fig.4?
- How can vaccination be taken into account in your study? The current situation is pretty different from 2020.
- The quality of the figures is to be improved.
I like this study, but I recommend refreshing it with new relevant data and clarifying the processes of data extraction and analysis. Nevetheless, I believe the paper can be accepted after only minor revisions.

Author Response

Thank you for your comments and suggestions. Please see the attachment for clarifications by the authors to reviewer 1.

Reviewer 2 Report

Dear authors,

You caught my attention with the abstract, but when I read the paper I found myself more confused than before reading.

Let me start from the most basic comments:

1) I needed to Google the words Jan Andolan and pandals because those words awere not described in the paper

2) you are using "Reported Test Positivity" in the sentence 155 and TPR in the formula, without ever mentioning that is the same

3) in the formula you are not using subscript for -1, hence rendering your formula unusable. I would prefer more effort and a somewhat background description of the methodology of chosen ATPR

4) in chapter 3.1 you are describing the recognition of two levels, with 3 individual CAB and five community CAB. I have numbered 7 community CABs in bullets? I would also prefer to see everything numbered (eg. 1a, 1b...)

5) all images are in low resolution making letters almost unreadable

6) I don't get how did you compiled figure 3 and what are the connections with the rest of the paper

7) Trendlines for the day-wise ATPR are not described, analysed or are adding much to the paper. I would consider ommiting ATPR from the final paper 

Author Response

The authors appreciate the valuable insights of the reviewer. Please see the attachment for clarifications to reviewer 2.

Reviewer 3 Report

Overall it is an interesting manuscript covering an interesting topic which is generally not explored well in published literature, so this is a good research article and direction. 

Authors need to make sure the referencing is consistent  in whole of the manuscript. Initially both number system and author system is used in the text (introduction) and then only number system in methods and discussion. Please follow IJERPH  referencing guidelines  

Author Response

Thank you for your valuable observation.

The referencing style has been thoroughly checked throughout the manuscript and the inconsistencies have been corrected. The changes are reflected in the new uploaded manuscript.

Round 2

Reviewer 2 Report

I still do not think that the whole TPR section is needed, but I could live with the decision to keep it in the paper. 

I do believe that some text editing is needed, some sentences are too long and I have noticed that some words are missing in a few places ( eg. in line 138 verb is missing)

Author Response

Thank you for indicating the inadvertently caused typological error. The missing word has been incorporated and the language has been edited.